# Optimised Stable Lighting Strengthens Circadian Clock Gene Rhythmicity in Equine Hair Follicles

**DOI:** 10.3390/ani13142335

**Published:** 2023-07-17

**Authors:** Aileen Collery, John A. Browne, Christiane O’Brien, John T. Sheridan, Barbara A. Murphy

**Affiliations:** 1School of Agriculture and Food Science, University College Dublin, Belfield, D04 V1W8 Dublin, Co. Dublin, Irelandjohn.a.browne@ucd.ie (J.A.B.); 2Equilume Ltd., W91 TP22 Naas, Co. Kildare, Ireland; 3School of Electrical, Electronic and Mechanical Engineering, University College Dublin, Belfield, D04 V1W8 Dublin, Co. Dublin, Ireland

**Keywords:** clock gene, hair follicle, horse, circadian, LED, blue light, red light, *PER2*, *NR1D2*

## Abstract

**Simple Summary:**

All mammals have an internal timing system that is responsible for regulating most aspects of physiology and behaviour. This internal timing system, or body clock, is regulated by the daily 24 h rhythms in light and dark exposure and functions in almost every tissue and organ. Domesticated species such as horses are often stabled and exposed to lighting at different times than they would in nature. The cells in hair follicles possess a clock that can be evaluated without the need for more invasive tissue collection. This study aimed to evaluate the clock in hair follicles of horses kept under two lighting systems. After 20 weeks housed with either an incandescent light bulb or a custom-designed LED lighting system, 24 h clock gene rhythms from hair follicles of horses housed under the LED lighting system were found to be stronger. The LED lighting system contained blue light by day, which is more like the light produced by the sun and is known to be responsible for keeping the biological clock ticking, and dim red light at night to help avoid the disruption caused by turning on a white light in the stables at night. Therefore, our results suggest that there is the potential to improve stable lighting for horses to optimise the function of the body clock and ultimately the health of horses. By improving the circadian (24 h) rhythm of horses, all aspects of their physiology can work in better harmony and in synchrony with the environment.

**Abstract:**

Hair follicles (HF) represent a useful tissue for monitoring the circadian clock in mammals. Irregular light exposure causes circadian disruption and represents a welfare concern for stabled horses. We aimed to evaluate the impact of two stable lighting regimes on circadian clock gene rhythmicity in HF from racehorses. Two groups of five Thoroughbred racehorses in training at a commercial racehorse yard were exposed to standard incandescent light or a customized LED lighting system. The control group received light from incandescent bulbs used according to standard yard practice. The treatment group received timed, blue-enriched white LED light by day and dim red LED light at night. On weeks 0 and 20, mane hairs were collected at 4 h intervals for 24 h. Samples were stored in RNAlater at −20 °C. RNA was isolated and samples interrogated by quantitative PCR for the core clock genes: *ARNTL*, *CRY1*, *PER1*, *PER2*, *NR1D2*, and the clock-controlled gene *DBP*. Cosinor analyses revealed 24 h rhythmicity for *NR1D2* and *PER2* and approached significance for *CRY1* (*p* = 0.013, *p* = 0.013, and *p* = 0.051, respectively) in week 20 in the treatment group only. No rhythmicity was detected in week 0 or in week 20 in the HF of control horses. Results suggest that lighting practices in racehorse stables may be improved to better stimulate optimum functioning of the circadian system.

## 1. Introduction

Regularly timed lighting that reflects daily light–dark fluctuations is an important synchroniser of an animal’s circadian system, an endogenous internal timing system that coordinates the function of each tissue in the body with the 24 h rhythms of the environment [1]. The strength (and appropriateness) of the environmental entrainment factors, primarily photoperiod, influences the robustness of molecular, cellular, and behavioural rhythms and ultimately affects overall health and welfare [2,3].

The mammalian circadian system is hierarchical with a central pacemaker residing in the suprachiasmatic nuclei (SCN) of the hypothalamus regulating the rhythms of tissue clocks throughout the body via humoral and neural mechanisms [4]. Peripheral tissue clocks, which take their cues from both photic and nonphotic entrainment factors, drive the circadian expression of gene subsets involved in their respective unique physiological functions [5]. Cells within the SCN ‘master’ clock, as well as peripheral tissue clocks located throughout the body, all possess a similar molecular clockwork mechanism [6,7,8]. It consists of a core set of clock genes that contribute to autoregulatory transcription–translation feedback loops supported by oscillating posttranslational modifications of proteins that provide rhythmic 24 h periodicity (see Albrecht, 2012, for review [2]). Studies in horses have identified oscillating core clock genes with temporal profiles similar to the molecular clockwork of other species in cultured fibroblasts, adipose tissue, blood, skeletal muscle, and hair follicles (HF) [9,10,11,12,13,14].

All light is not equal in its ability to entrain the circadian system. Short-wavelength blue light optimally stimulates a set of novel photoreceptors in the mammalian retina that regulates the master circadian pacemaker in the SCN [15]. These intrinsically photosensitive retinal ganglion cells (ipRGCs) relay ‘time-of-day’ messages along the retinohypothalamic tract (RHT) to the SCN [16,17], and from there the signal is transmitted throughout the body. Projections from the SCN to the pineal gland modulate daily and seasonal changes in melatonin secretion [18], an important mediator of photoentrainment for the circadian system [19]. Recent studies in horses have demonstrated a dose dependent suppression of melatonin by blue light and the effectiveness of low-intensity blue light for advancing seasonal reproductive cycles in mares [20,21].

In contrast, long-wavelength red light has a reduced capacity to stimulate ipRGCs [22]. As ipRGCs are fundamentally involved in light-induced melatonin suppression via the RHT [23], avoiding their stimulation at night has important implications for nighttime rest and proper circadian functioning. In rodents, white light intensities as low as 5 lux have been shown to disrupt the sleep–wake cycle [24], whereas ≤10 lux of red light facilitated normal activity–rest behaviour and metabolic function [25,26]. In humans, red vizors have been tested for nighttime use in shift workers and shown to help reduce melatonin suppression [27]. Importantly, dim red stable lighting was recently shown to permit the normal nocturnal rise in melatonin in horses [28].

Disturbance of SCN signalling to peripheral clocks leads to arrhythmicity of clock gene expression and circadian dysfunction and can be caused by SCN lesions or prolonged exposure to constant or irregular light [29,30,31]. The development of practical methods to assess the circadian system and, thus, detect circadian disruptions and/or the circadian phase is an important area of human medical research. Common circadian biomarkers used in both human and equine studies include monitoring serum or salivary melatonin, core body temperature, or tissue collection for the assessment of clock gene profiles [10,22,23,24,25,26,27,28,29,30,31,32,33,34,35,36], but all have significant practical drawbacks. The identification of cycling clock genes in hair follicles (HF) provided a less invasive means of assessing internal circadian synchrony in humans [37], and was also confirmed as effective in horses [11]. The widespread usefulness of HF as a model for monitoring the central circadian clock was recently reviewed [38].

Racehorses spend much of the 24 h day indoors and unexposed to optimal daily light wavelengths. These horses are also frequently disturbed by light at night for feeding or nighttime checks. We hypothesized that this typical erratic exposure to light potentially contributes to reduced circadian rhythmicity. To evaluate this, we investigated 24 h clock gene expression profiles in HF from horses exposed to a standard stable lighting regime using incandescent light and a customised LED lighting system utilising blue-enriched, polychromatic white light by day and dim red light at night.

## 2. Materials and Methods

### 2.1. Animals

This study was approved by University College Dublin’s Animal Research Ethics Committee (Reference number: AREC-15-45-Murphy) and licensed by the Health Products Regulatory Authority of Ireland in accordance with the European Community Directive 86/609/EC.

Ten clinically healthy Thoroughbred horses (aged 3–4 years) were used in a randomised controlled trial at a single flat racing yard located in Co. Kildare, Ireland (latitude 53.17 and longitude −6.89) between January and June. All horses were housed in individual stables (3.7 m × 4.3 m) and exercised for a period of one hour, six days per week between 08:00 and 12:00 h for the duration of the study. Horses had access to ad lib hay and water and were supplemented daily with between 7.5 and 9 kg of concentrated feed (Connolly’s Red Mills Sweetfeed Racehorse Mix 12.5%) depending on individual animal requirements as assessed by the trainer. Rations were delivered four times daily at 05:00, 11:30, 17:30, and 23:00 h. Prior to study commencement, stable lighting was provided to all horses via traditional light fittings consisting of 60-Watt incandescent light bulbs (E27 GLS, Phillips, Amsterdam, The Netherlands) contained within opaque plastic fittings. Natural light was provided via open top doors and roof skylights in each stable. The daily lighting regime consisted of switching lights on at the time of morning feeding (05:00 h) until there was sufficient natural light for staff to clean stalls and tack up and then lights were switched off. Lights were switched on in the late afternoon until 18:00 h and then switched off. Lights were turned on briefly at 23:00 h to facilitate feeding. This regime was not strictly adhered to and interactions with horses at other times necessitated switching lights on at irregular times, as is typical of racehorse management. The experimental design (described below) was such that all horses were exposed equally to the same ambient temperature changes as they were all housed in the same yard at the same GPS coordinates for the entire 20-week study period.

### 2.2. Experimental Protocol and Lighting

On week 0, five horses were randomly selected for the collection of baseline samples. Mane hair samples were collected at 4 h intervals from 08:30 h for a period of 24 h. Sampling during darkness hours was conducted with the aid of small red headlamps. Between 15 and 20 hairs, complete with hair follicles, were carefully plucked, trimmed, and stored in 2 mL tubes containing 400 µL RNAlater (Thermo Fisher Scientific, Waltham, MA, USA). Samples were maintained at 4 °C for a period of 24 h and at −20 °C thereafter.

Following baseline sampling, horses were blocked for gender and assigned to a treatment *(n =* 5) or a control (*n* = 5) group. Each group consisted of four fillies and one colt. The standard daily lighting regime was maintained for the control group and the staff were advised to continue to use lights as was normal for the yard. The treatment group had a customised lighting system installed in each stable with the following features: Daytime polychromatic light was supplied by a fixture containing white LEDs (peak wavelength 455 nm at absolute irradiance of <0.84 μW/cm^2^/nm). Supplemental light was provided via four blue LED bulbs (450 nm at absolute irradiance of <0.6 μW/cm^2^/nm) mounted on a rectangular frame surrounding the white light fixture at 3.5 m from floor level. Each blue LED bulb provided 100 lux light intensity at 1 m below light source. Combined, the customised lighting provided an average 300 lux at horse eye level within the stable. Dawn and dusk were simulated by gradual increases in light intensity from 5 lux to ~300 lux and gradual decreases from ~300 lux to 5 lux, respectively, over 20 min intervals. An automatic timer controlled the duration of daily exposure to light such that dawn occurred at 05:40 and dusk occurred two hours after natural environmental dusk. The lighting dimmed to red light at night (5 lux, peak wavelength 625 nm at absolute irradiance of <0.3 μW/cm^2^/nm), at an intensity and wavelength previously shown to permit the normal nocturnal melatonin rise in horses [25]. The light intensity and spectral composition of treatment and control lighting were recorded using the CL-70F Illuminance meter (Konica Minolta). The treatment lighting instalment and respective light spectra for white, red, and blue lights, as well as the control lighting spectrum, are depicted in Figure 1. On week 20, following daily exposure to the control or treatment light regime, hair samples were collected at 4 h intervals from 17:00 h for a period of 24 h, as described.

### 2.3. Real-Time Quantitative Polymerase Chain Reaction (qPCR)

Total RNA was isolated from hair follicle samples using the High Pure RNA Isolation kit (Roche Life Sciences, Indianapolis, IN, USA) as previously described [11] with minor modifications. Hair follicles, previously stored in RNAlater, were first transferred to a 1.5 mL microcentrifuge tube containing 400 µL of Lysis/Binding Buffer and 200 µL of PBS, with care taken to remove as much as possible of the remaining hair shaft without losing the follicle. All RNA samples were of good quality and shown to have a RIN value greater than 7.5. For each sample, 200 ng of total RNA was converted to cDNA using the High-Capacity cDNA Reverse Transcription Kit (Thermo Fisher) as previously described [11].

Real-time quantitative (q) PCR assays were performed using Biosystems 7500 Sequence Detection System and SYBR Green Master mix (Bioline, London, UK). A panel of eight potential reference genes (Table 1) were assessed for stability using the GeNorm algorithm within the qBase+ qPCR analysis software package (Biogazelle, Gent, Belgium). GeNorm analysis identified *GAPDH* and *PPIA* as the most stably expressed reference genes (M < 0.25) and the geometric mean of these two genes was used to normalise the qPCR expression data within the qBase+ package. All qPCR reactions were prepared as previously described [11]. A panel of five core clock genes were selected: *PER1* (Period homolog 1), *PER2* (Period homolog 2), *ARNTL* (Aryl hydrocarbon receptor nuclear translocator-like), *CRY1* (Cryptochrome 1(photolyase-like), *NR1D2* (Nuclear receptor subfamily1, group D, member 2), and one clock output gene *DBP* (D-site of albumin promoter binding protein). *DBP* is important for its role in the circadian oscillatory mechanism and as a clock-controlled gene that is robustly rhythmical in most tissues [39,40]. These candidate genes were chosen based on previous evidence of their expression in equine hair follicles [11] and of their cyclic expression in human hair follicles [37]. Previously designed primer sequences were commercially synthesised using Eurofins MWG Operon [10] (Table 2).

### 2.4. Data Analysis

Gene expression data were normalised relative to the highest values to account for differences in the relative abundance of gene transcripts between individuals. The presence of 24 h temporal variation in transcript means was evaluated using the Cosinor programme [41] based on the least squares cosine fit method [42] available at https://www.circadian.org/softwar.html (accessed on 1 July 2023) This cosinor method also computed an estimate of acrophase (time of peak value of the fitted cosine function) and robustness (measure of rhythm stability). Significant 24 h variation was assessed as *p* < 0.05. Tests for normality were run and graphs created using Prism 9 for MacOS. Data are presented as means ± SEM.

## 3. Results

Transcripts for all candidate genes were detected in HF cells at all sampled times. Normal distribution of the data was confirmed using the Kolmogorov–Smirnov test for normality (*p* > 0.05). No 24 h temporal expression patterns were detected in baseline samples taken on week 0 under control lighting conditions (Figure 2). On week 20, statistically significant 24 h expression patterns were detected for *NR1D2* and *PER2* and approached significance for *CRY1* (*p* = 0.013, *p* = 0.013, and *p* = 0.051, respectively) in the treatment group only (Figure 3). No significant rhythmical patterns were detected in the control group (Figure 3). Cosinor results are presented in Table 3.

## 4. Discussion

This study aimed to determine circadian rhythmicity in a peripheral tissue of racehorses maintained under two different lighting regimes by assessing core clock gene expression patterns in HF. HF have been described as a useful, noninvasive source tissue for monitoring the central circadian clock in humans [38]. The oscillating expression of clock genes in HF has previously been demonstrated in horses maintained under a light–dark cycle [11]. In this study, horses were maintained under a typical daily lighting regime for racehorse stables using incandescent bulbs or a customised, controlled LED lighting system. We showed, that under noncontrolled incandescent light exposure, horses did not demonstrate rhythmical clock gene expression in this peripheral tissue but that rhythmicity in specific clock genes was apparent after 20 weeks under an LED lighting system incorporating blue-enriched polychromatic light by day and dim red light at night.

We surmise from these results that the treatment lighting provided a strong entrainment signal to the circadian system. To allow coherence between internal and external timing, environmental entraining signals, primarily photoperiod, directly synchronise the master circadian clock in the SCN. The SCN in turn synchronises oscillators in peripheral tissues through numerous pathways that include neural, blood-borne, feeding-fasting, and body temperature signals [19,43,44,45]. A conserved molecular clock mechanism exists in SCN and peripheral cells and consists of transcription–translation feedback loops [1] comprising a group of clock genes whose roles have been comprehensively described [2]. Synchronised activity of cellular clocks permits downstream activation of clock output genes, such as DBP [46], that in turn activate clock-controlled genes with tissue specificity that allows for individual tissue functionality [47,48].

We speculate that failure to detect clock gene oscillations in hair follicles on week 0, and in the control group on week 20, is due to the absence of a sufficiently effective entrainment cue that results in dampened or desynchronised rhythms in this tissue. The first and last samples for some clock genes (taken in the same 24 h clock time) do not show the same mean relative expression level and this is indicative of a non-24-hour oscillation suggesting a nonsynchronised peripheral clock. However, in the case of DBP, NR1D2, and PER1 in the control group on week 20, outside of differences in individual variation, it is unclear why the last sample appears to show lower mean transcript abundance levels than the first sample. Whereas individual neurons in the SCN are capable of maintaining circadian phase coherence in the absence of environmental timing signals [49,50], peripheral tissue clocks rely on daily internal timing signals to maintain rhythm coherence between cells [51]. In the absence of entraining signals, circadian rhythms free-run with a period close to 24 h and this manifests as dampened or desynchronised rhythms in peripheral tissues [6], reflecting a gradual desynchrony of many independent cellular oscillators [51].

Different peripheral clocks are sensitive to synchronisation cues from various entrainment factors. Exercise, for example, is a potent entrainment cue for skeletal muscle [52,53] and regularly timed daily exercise was shown to synchronise circadian gene expression in equine skeletal muscle [54]. In peripheral tissues with primary metabolic functions, the daily fluctuations in energy availability provide the dominant entrainment cue [55]. In fact, feeding–fasting rhythms are thought to be the dominant phase-resetting cue for most peripheral organs [56].

In the current study, all horses experienced the same regularly timed feeding and exercise schedule. The only circadian entrainment cue that varied was the quality and consistency of the environmental lighting signal. As blue light optimally stimulates ipRGCs [15], the emergence of oscillating 24 h clock gene expression patterns in hair follicle cells in the treatment group alone may suggest that light signals mediate entrainment of the molecular clock in this tissue and that the signal relies on ipRGC neurotransmission. This supports an important finding in mice showing that ipRGCs are necessary for light entrainment of peripheral clocks [57]. More recently, blue light stimulation was shown to cause rapid activation of hair follicle stem cells via an ipRGC-SCN-sympathetic nervous circuit in mice [58], providing a plausible pathway for synchronisation of this peripheral clock.

Alternatively, melatonin, an important mediator of photoentrainment for the circadian system [19] is a likely candidate for transmission of the signal from the SCN to hair follicles. Polychromatic, blue-enriched light has been shown to elicit greater daytime melatonin suppression over standard white light in humans [59]. Research conducted within an environment where ambient light intensity was chronically low to moderate (polar base station in Antarctica) found that artificial, blue-enriched light alleviated disruptions in the circadian phase of melatonin production, as well as improving cognitive performance and mood [60]. However, a limitation of the current study is that it is not possible to determine whether it was the additional exposure to blue-enriched light by day, the elimination of white light exposure at night facilitated by the dim red nighttime light, or the combined effect of both that was responsible for the strengthened circadian output observed in the treatment group. It is worth noting that both treatment and control stables contained ceiling skylights (evident in Figure 1A) permitting some natural light exposure within the stables by day; however, the intensity was weather dependent and variable. Nevertheless, it may be that the regular transition to dim red light each night and the associated uninterrupted period of melatonin production [28] may have created a stronger entrainment cue to the circadian system. This is especially relevant given that horses are acutely sensitive to white light stimulation during the hours of darkness and an immediate drop in plasma melatonin concentrations accompanies the switching on of stable lights at night [34].

In contrast, the use of incandescent light bulbs that emit minimally in the short wavelength portion of the spectrum [61], in addition to white light disruption during nighttime interventions, may explain the absence of clock gene oscillations in control horses. While the exact nature of the entrainment mechanism in treatment horses is unknown, we hypothesize that the light-mediated synchronising signal from the SCN to HF occurred either directly, via an ipRGC-SCN-neural circuit, or indirectly, via melatonin signalling. Future studies should evaluate the capacity of a strengthened photic entrainment cue to synchronise rhythms in equine tissues after shorter exposure periods.

Circadian clock genes participate in the regulation of several physiological processes in HF as well as being synchronised by the central circadian clock [38]. Hair follicles progress continuously through a hair growth cycle that has three main phases, anagen (active growth), catagen (involution), and telogen (quiescence), and requires a time frame for cycle completion that varies from weeks to months [62]. Recently, it was shown that circadian clock genes regulate anagen progression via their effect on the cell cycle and it was speculated that circadian control mechanisms for hair follicles evolved to allow seasonal regulation of hair growth [63]. Indeed, a role for ipRGC signalling in response to blue light in the regulation of seasonal moulting in horses was recently highlighted [64]. There is also evidence that enhanced core clock gene expression may occur at different phases of the hair growth cycle [63]. Thus, it is feasible that, after 20 weeks under two different lighting regimes, the groups may have differed in the phase of their respective hair growth cycles.

Investigation of clock gene expression in human HF has revealed that hair thickness and individual variation influences transcript abundance [37]. The authors also reported only a slight oscillation for *ARNTL*. Similarly, we found large variations in transcript abundance between individual horses, potentially explaining the failure to find significant rhythmicity for certain genes, and *ARNTL* also showed no 24 h pattern in treatment horses.

Another explanation for the lack of temporal oscillation in some clock genes on week 20 is the known existence of functionally redundant clock gene isoforms in certain tissues [65]. Nevertheless, our finding of a high-fit cosine curve oscillation for *NR1D2* in this and a previous study [11] agrees with reproducible findings for this transcript in human hair follicles [37] and supports its use as a circadian rhythm marker in the horse. Moreover, *PER2* may equally serve as a rhythm marker for equine peripheral clocks as a 24 h temporal expression pattern was also previously detected in skeletal muscle [10] and adipose tissue [9].

The concept of circadian lighting as it relates to human health stems from our understanding of the beneficial impact of blue light exposure by day and its chronodisruptive effect by night [66,67]. Our ability to provide environments that support internal circadian cohesion, maintaining synchrony between the SCN and peripheral oscillators, could have widespread benefits for equine health and performance [68]. Future investigations should evaluate the impact of custom LED lighting regimes, such as that described here, on health and behavioural parameters in the horse.

## 5. Conclusions

The results of this study suggest that standard stable lighting regimes used in racehorse management may be inadequate for the optimum synchronisation of the circadian system. This study also provides the first evidence that a customised LED lighting system can influence clock gene rhythms in the peripheral tissue of the horse. The evaluation of clock gene oscillation in HF may serve as a useful biomarker for estimating circadian functionality in horses under diverse management conditions.

## Figures and Tables

**Figure 1 animals-13-02335-f001:**
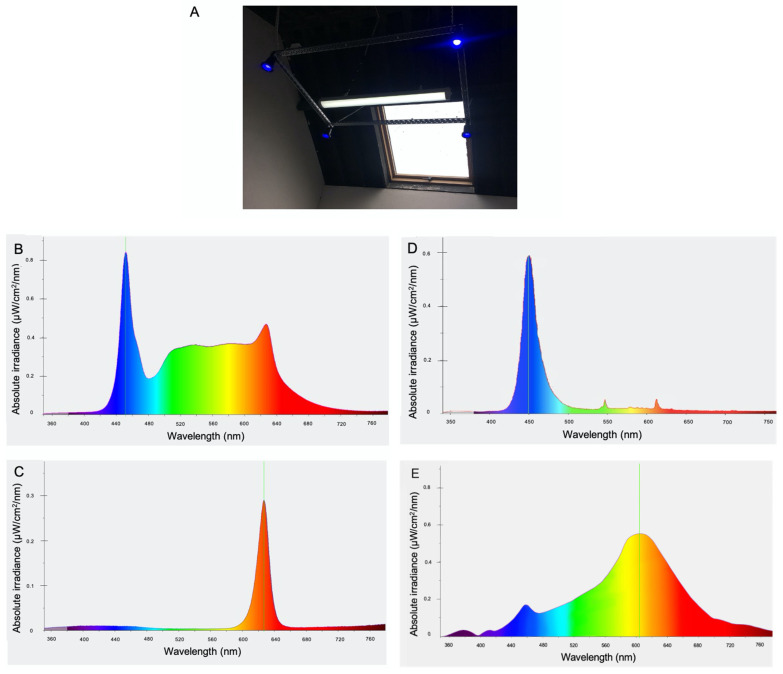
(**A**) Image showing treatment lighting instalment consisting of white LED fixture (including red LEDs for activation at night) surrounded by four blue LED lamps mounted on a rectangular frame suspended at 3.5 m from floor level. Light spectrum for polychromatic white light (**B**), red light (**C**), supplemental blue light (**D**), and light spectrum provided by incandescent bulb used in control stables (**E**).

**Figure 2 animals-13-02335-f002:**
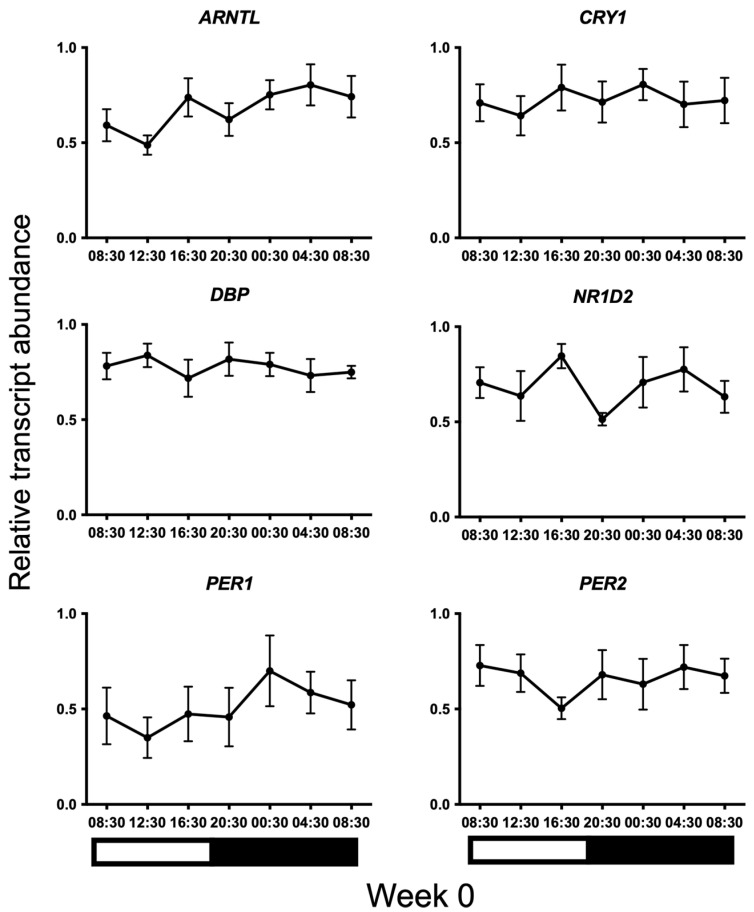
24 h profiles of mRNA expression on baseline Week 0 under control lighting conditions. Plotted are normalised mRNA levels of the candidate genes relative to reference genes GAPDH and PPIA in equine hair follicles. Data are presented as means ± SEM. Cosinor analyses revealed no significant 24 h variation for any gene (*p* > 0.05). Light and dark bars represent the light–dark cycle associated with the natural environmental photoperiod during sample collection.

**Figure 3 animals-13-02335-f003:**
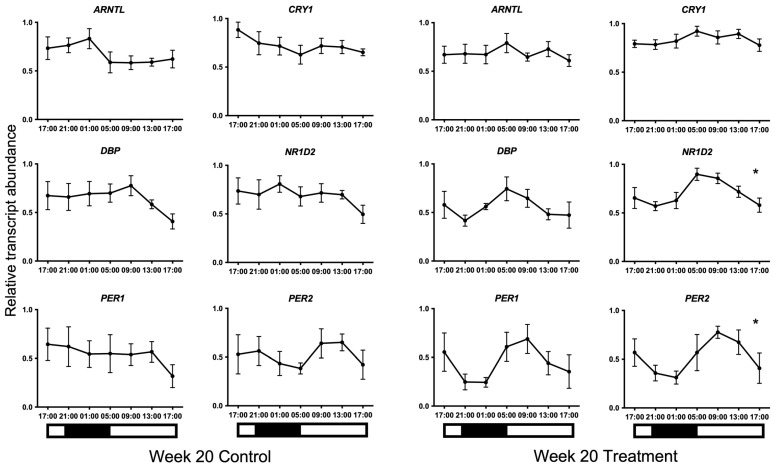
24 h profiles of mRNA expression on Week 20. Plotted are normalised mRNA levels of the candidate genes relative to reference genes *GAPDH* and *PPIA* in equine hair follicles in the Control and Treatment lighting groups. Data are presented as means ± SEM. Cosinor analyses revealed significant 24 h variation for *NR1D2* and *PER2* and approached significance for *CRY1* (*p* = 0.013, *p* = 0.013, and *p* = 0.051, respectively) on Week 20 in the treatment group only. * denotes significance at *p* < 0.05. Light and dark bars represent the light–dark cycle associated with the natural environmental photoperiod during sample collection.

**Table 1 animals-13-02335-t001:** Reference genes assessed for suitability in qPCR assays of mRNA from equine HF cells.

Gene Symbol	Forward Primer Sequence (5′-3′)	Reverse Primer Sequence (5′-3′)
*ACTB*	CACCTTCTACAACGAGCTGC	CGGGGTGTTGAAGGTCTCA
*GAPDH*	GGAGTCCACTGGTGTCTTCA	GTTCACGCCCATCACAAACA
*H3F3A*	CAAACTTCCCTTCCAGCGTC	TGGATAGCACACAGGTTGGT
*PPIA*	GCATCTTGTCCATGGCGAAT	CAAAGACCACATGCTTGCCA
*RPL19*	CTGATCATCCGGAAGCCTGT	GGCAGTACCCTTTCGCTTAC
*SDHA*	TGTTGTGTCTCGGTCCATGA	AGATCATGGCCGTCTCTGAA
*YWHAZ*	AGACGGAAGGTGCTGAGAAA	CTTGTGAAGCATTGGGGATCA
*RNF11*	AGGATAGCTCAAAGAATAGGC	CGGCAGAAATCGAATTGGGT

**Table 2 animals-13-02335-t002:** Quantitative PCR primer sequences for equine clock genes.

Gene Symbol	Forward Primer Sequence (5′-3′)	Reverse Primer Sequence (5′-3′)
*ARNTL*	CACCTTCTACAACGAGCTGC	CGGGGTGTTGAAGGTCTCA
*CRY1*	GGAGTCCACTGGTGTCTTCA	GTTCACGCCCATCACAAACA
*DBP*	CAAACTTCCCTTCCAGCGTC	TGGATAGCACACAGGTTGGT
*NR1D2*	GCATCTTGTCCATGGCGAAT	CAAAGACCACATGCTTGCCA
*PER1*	CTGATCATCCGGAAGCCTGT	GGCAGTACCCTTTCGCTTAC
*PER2*	TGTTGTGTCTCGGTCCATGA	AGATCATGGCCGTCTCTGAA

**Table 3 animals-13-02335-t003:** Results of cosinor analyses of mRNA expression profiles. For each gene transcript the robustness values, acrophase time, and *p* value are presented. * denotes significance at *p* < 0.05. Acrophase times are only indicated where *p* < 0.1.

	Week 0			Week 20 (Control)			Week 20 (Treatment)		
Gene Transcript	Robustness (%)	Acrophase (24 h)	*p* Value	Robustness (%)	Acrophase (24 h)	*p* Value	Robustness (%)	Acrophase (24 h)	*p* Value
*ARNTL*	36.0	n/a	0.411	74.1	22:31	0.068	26.0	n/a	0.550
*CRY1*	36.9	n/a	0.399	24.2	n/a	0.577	77.8	07:45	0.051
*DBP*	6.0	n/a	0.882	44.0	n/a	0.313	63.5	n/a	0.134
*NR1D2*	1.6	n/a	0.970	2.5	n/a	0.603	89.5 *	07:47	0.013
*PER1*	68.2	n/a	0.101	2.6	n/a	0.949	65.3	n/a	0.121
*PER2*	48.2	n/a	0.268	37.6	n/a	0.391	89.6 *	10:23	0.013

## Data Availability

Data available on request from the corresponding author.

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
