# Peer review of "Optimised Stable Lighting Strengthens Circadian Clock Gene Rhythmicity in Equine Hair Follicles"

_animals, 2023, doi:10.3390/ani13142335_

Round 1
Reviewer 1 Report
The study investigated the change of several clock gene and a clock controlled gene (DBP) expression in hair follicle cells of horses exposed to two lighting regimes. The study is interesting and it was well written, however it has several critical points.
Critical points:
1) Incandescent light bulbs were used for the control group. However, incandescent bulbs are no longer used for lighting as their use has progressively switched to the LED lamp.
2) Several variables could interfere with the results, therefore it is not possible to understand the exact causal effect. It has been partially explained in the text as limitations of the study.
3) The lock gene expression level in the first sample compared to the final sample had different levels. This was strange and suggested a non-24 hour oscillation period or problems with sampling or laboratory procedures.
4) The logic of investigation of DPB gene is not explained.
5) The light output spectrum of the bulb used in the control group is not shown.
6) Possible ambient temperature difference is a major confounder and was not considered in the study.
7) The number of horses is very low and interindividual differences can play an important role in the results obtained.
Reviewer 2 Report
The study is original and worth publishing after minor correction. It is a pity that there were only 5 horses in each of two groups studied. It would be much better if there were six. Were there any differences between fillies and colts?
Both summaries are not fully understandable with regard to the division into groups.
“…horses kept under two lighting systems.” What systems? This information is not clear without specifying the systems. And again “…to better replicate..” Better than what?
The same remark to the Abstract. “…two stable lighting regimes” should be specified instead of writing on the temperature of storing the samples, number of follicles, healthy horses (this is obvious and not necessary in abstract). At first glance, I thought there were two tested groups and one control group. It should be written that there were two groups of five horses tested: a treatment group and a control group.
Instead of writing that “analyses approached significance” I would write that the difference between week 0 and week 20 approached significance for… Perhaps the values of p are redundant.
Materials and Methods
I would not write the study was performed in 2017.
2.2. chapter
It should be written shorter. My suggestion is to firstly describe the division of horses, then the systems of lightening and next the collection of samples.
Reviewer 3 Report
The aim of this study was to evaluate the impact of two different stable lighting regimes on circadian clock gene expression in hair follicles of racehorses.
During the last decade, the extraction of RNA from equine hair follicle cells has been reported to identify the circadian oscillations of specific clock genes, providing a valuable non-invasive method for evaluating the equine peripheral circadian clock.
Overall, this is an interesting and well-written study, highlighting the importance of lighting regimes in racehorse stables to stimulate optimum functioning.
I have some minor comments for improvements.
Lines 102 and 127: "Ten healthy Thoroughbred horses". Please elaborate the term "healthy". Are there any lab tests performed or the term used is for clinically healthy horses? The genes can be modulated by both the central circadian system and some other factors, such as thyroid hormones. Did the authors perform any additional lab panels including parameters that may affect hair growth e.g. thyroid function, cortisol concentration etc?
Racehorses are often treated with steroids. Have the horses included in this study received any IA or other injections during the 6-month period? If not I would suggest to add these details in the Materials and Methods section.
Additionally, in the materials and methods section the authors comment that following baseline sampling, horses were blocked for gender and assigned to groups. I would suggest a short comment regarding the selection process. For example in previous studies geldings were generally considered unsuitable as their neuroendocrine system is often compromised.
Line 205: Discussion section although containing useful bibliography, in some points, it is lacking clear conclusions regarding the specific study. For example section from line 216-234. I would suggest clear transition sentences that exemplify the authors' notion, followed by the relevant bibliography.
Round 2
Reviewer 1 Report
The quality of manuscript has been partially improved, but few information has been added or discussed in the text (e.g. the specification of no differences in ambient temperature between groups should be added, the possible reasons for differences of first and final sample on the baseline week 0, and week 20 for the control group should be discussed, the inclusion of the spectrum of incandescent bulbs in Figure 1 should be cited in the text).
The last sentence of “Simple summary” is speculative and not supported in the text.
The “Cosinor programme” used for statistical analysis should be specified.
The instrument used for spectrum and absolute irradiance definition should be specified. The spectrum of incandescent bulbs used for the control group and specified in the new Figure 1E may refer to a fluorescent bulb. The specific manufacturer and model of incandescent bulb to which the spectrum in Figure 1E refers is important and should be included in the text.
